# Principles of Imitation for the Loading of the Test Bench for Gas Turbines of Gas Pumping Units, Adequate to Real Conditions

Anton Petrochenkov [1,*], Aleksandr Romodin [1], Vladimir Kazantsev [1], Aleksey Sal'nikov [1], Sergey Bochkarev [1], Yuri Gagarin [2], Ruslan Shapranov [2] and Pavel Brusnitcin [2]

[1]  Electrical Engineering Faculty, Perm National Research Polytechnic University, 614990 Perm, Russia; romodin@mail.ru (A.R.); zav@msa.pstu.ru (V.K.); afsalnikov_1@mail.ru (A.S.); bochkarev@msa.pstu.ru (S.B.)

[2]  Technical Department, Sputnic-Komplektatsia LLC, 614990 Perm, Russia; gua@sputnic.ru (Y.G.); r.shapranov@sputnic.ru (R.S.); p.brusnitsyn@sputnic.ru (P.B.)

*  Correspondence: pab@msa.pstu.ac.ru; Tel.: +7-3422-391-821

**Abstract:** The purpose of the study is to analyze the prospects for the development of loading methods for gas turbines as well as to develop a mathematical model that adequately describes the real operating conditions of the loading system at various loads and rotation speeds. A comparative analysis of the most common methods and technical means of loading the shafts of a free turbine at gas turbine plants intended for operation as part of gas pumping units is presented. Based on the results of the analysis, the expediency of using the loading model "Free Power Turbine Rotor–Hydraulic Brake" as a load simulation is shown. Recommendations for the creation of an automation system for the load testing of power plants have been developed. Mathematical models and Hardware-in-the-Loop simulation models of power plants have been developed and tested. One of the most important factors that predetermine the effectiveness of the loading principle is the possibility of software implementation of the loading means using software control systems that provide the specified loading parameters of the gas turbine.

**Keywords:** imitation the loading; loading device; test bench; gas turbine engine; gas turbine unit; gas pumping unit; mathematical modeling

## 1. Introduction

Minimizing the negative impact of energy systems on the environment, as well as reducing the technogenic impact on the ecosystem, are two of the key tasks on today's agenda. The improvement of gas transportation technologies is an integrated task—from the production of power plants to the development of advanced approaches to diagnostics, monitoring, and the control of test benches of power plants.

Field tests of gas turbine engines (GTE) of various manufacturers are characterized by multifactorial and multipurpose studies.

A specific feature of the testing technology of gas turbine units (GTU) of gas pumping units (GPU) is the creation of a controlled load torque on the shaft of a free turbine (FT) using a special loading unit—a loading device. This requires a significant investment of time and effort to prepare for testing—installation, alignment, preparation of test programs, calibration of sensors, etc. [1]. Hence, the issues of justifying the choice of the principle and the technical means of loading the gas turbine, as well as the control system of the storage device and measuring the parameters of the gas turbine during load tests, become very relevant.

Gas path diagnostics is an integral part of condition-based maintenance of gas turbines. In addition, the literature on the diagnostics of gas turbines is known, and many methods have been proposed. However, the fundamental limitations of traditional methods are the inability to cope with the nonlinear behavior of the motor, measurement uncertainty, simultaneous malfunctions, and a limited number of measurements on the bench [2–4].

Loading testing of gas turbine units presupposes the presence of fully functional test benches [5,6], providing modes of manual and automatic programmed loading of the GTU's free turbine shafts, as well as providing all the necessary conditions for control and acceptance tests of gas turbine units. In addition to fully functional test benches, benches for semi-natural testing (HIL-modeling) [2,7,8], as well as benches for testing gas turbine engines under partial load and various laboratory benches (including educational research benches), are used.

Fully functional benches are used, among other things, for engines converted for the needs of pumping gas through main gas pipelines and for the needs of small-scale power generation [5,9].

## 2. Overview of Methods of Loading the Test Bench

A gas turbine consists of complex, very expensive, and precise components that operate under conditions of high gas pressure and temperature. Engine failure or performance degradation greatly affects engine performance. Component performance may deteriorate due to various malfunctions such as clogging, erosion, blade gap, corrosion, and object damage. A detailed description of these faults and their effects on the performance of the gas turbine, including the corresponding recommendations for maintenance, can be found in the study carried out in article [10].

There is a lot of research published in the field of gas turbine diagnostics, focusing on component characterization. For example, article [11] presented an improved scheme and diagnosis of gas path faults for stationary gas turbines with a single spool, which can simultaneously approximate error components and sensors. Similar to this approach, article [12] presents an algorithm for diagnosing the components of the gas path of a gas turbine engine, which can isolate single, double, and multicomponent faults in the presence of noisy data. More details about this can be found in article [13]. However, recent studies in articles [14–16] show that existing algorithms still need to be improved, and there is some scope for further study in this area.

In general purposes, computerized model for diagnosis and prediction of the gas turbine performance are developed in a long period of time. Using a reliable, productive and adaptable maintenance method significantly improves the quality of operation and reliability of the gas turbine and thus minimizes unexpected failures, downtime and operating costs. Article [17] presents a review of published works on diagnostic strategies of gas turbines.

Article [18] is devoted to the analysis of failures of the compressor disk of an aircraft gas turbine engine made of titanium alloy based on the principles of simulation. Failures were due to partial surface cracks in the disk and dovetail blade mount. Based on the dimensions of this attachment, two geometries of simulation models of the compressor disk of a gas turbine engine have been developed. Accordingly, the loading conditions of the simulation models are found and numerically verified to reproduce the loading conditions during operation.

Thus, current approaches to managing the conditions of gas turbine engines are aimed at obtaining accurate information about the status condition of gas turbine engine components to optimize maintenance solutions in terms of economy and safety. Thus, in article [19], performance indicators based on physical characteristics are determined on the basis of deviations of the measured performance parameters compared to the predictions of the corresponding model. The indicators allow for effective monitoring of the deterioration of the characteristics of a gas turbine engine in both short-term and long-term modes. The developed methods and schemes are validated using a three-year performance dataset of an industrial gas turbine engine at a power plant.

Articles [1,20,21] mention the importance of estimating eigenfrequencies when the bench is operating. The proposed solutions have general applicability and can be extended to many other different cases.

Article [22] notes the importance of evaluating the test program on the bench, revising it with a simulation model to take into account specific factors.

The most widespread in the practice of constructing fully functional benches for testing a gas turbine under load are two models of loading FT shafts. The first is based on the application of the principle of electromechanical loading; the second is based on hydraulic loading.

### 2.1. Electromechanical Loading Models

Electromechanical loading of a GTU's free turbine is organically inherent in GTUs operating as part of gas turbine power plants, since the FT shaft is kinematically connected to the shaft of a synchronous generator (SG) [7,23]. In this case, the stator winding of the SG, directly or through an isolation transformer, is connected to the load resistors.

Let us consider several such schemes.

2.1.1. Loading Model According to the System "Free Power Turbine Rotor—Speed Reducer—DC Generator—Load Resistor"

The scheme of such a loading system is shown in Figure 1.

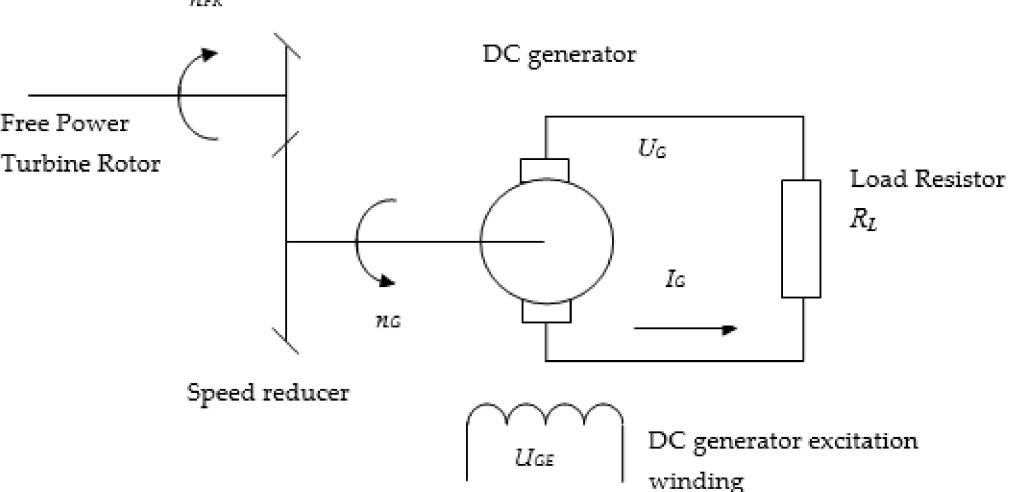

**Figure 1.** Scheme of the loading model "Free Power Turbine Rotor—Speed reducer—DC generator—Load resistor".

Designations on this scheme are: $n_{FR}$—rotation speed of the free turbine shaft; $n_G$—speed of rotation of the rotor of the DC generator; $U_{GE}$—DC generator excitation voltage; $I_G$—current flowing in the "winding of the generator—load resistor" circuit.

The control of the current in the load in such a scheme is implemented through the excitation circuit of the generator. In this case, the following relations take place:

$$I_G = U_G/R_L, \tag{1}$$

where $R_L$—the active resistance of the load resistor (neglecting the resistance of the brush contact and the lead wires due to their smallness in comparison with the resistance of the load resistor); $U_G$—voltage at the pins of the generator anchor winding,

$$U_G = E_G + R_S \cdot I_G, \tag{2}$$

where $E_G$—electromotive force generated in the generator anchor winding,

$$E_G = 2\pi \cdot n_G \cdot C_{CG} \cdot \Phi_G/60, \tag{3}$$

where $C_{CG}$—generator's constructive constant; $R_S$—equivalent resistance of the generator anchor winding; $\Phi_G$—magnetic flux created by the excitation winding.

Thyristor exciters [24,25], which have low control inertia and high linearity of the static characteristic, are usually used as a power-controlled converter to power the field winding. In this regard, the output voltage of the exciter is assumed to be proportional to the control voltage (or current) $U_{EC}$, i.e.,

$$U_{GE} = K_{TE} \cdot U_{EC}, \tag{4}$$

where $K_{TE}$—transfer coefficient thyristor exciter.

For generators of high power, it is necessary to take into account the influence of whirling currents on the dynamics of the excitation control loop, which will affect the increase in the equivalent time constant of the generator by the value of the time constant of whirling currents.

### 2.1.2. Loading Model According to the System "Free Power Turbine Rotor—Speed Reducer—DC Generator—Voltage Invertor (Current Invertor)—Supply Network"

More promising in terms of energy utilization seems to be another model based on the use of a controlled power converter that performs the function of controlled recovery of DC energy in the load of a DC generator into AC power at industrial frequency and operates on an industrial three-phase AC network. The scheme of such a loading system is shown in Figure 2.

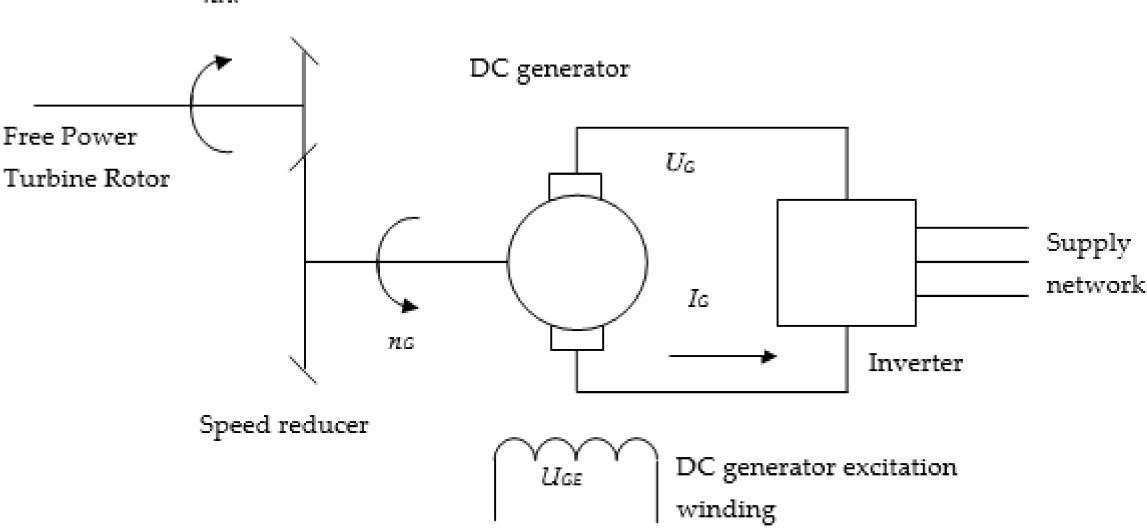

**Figure 2.** Scheme of the loading model "Free Power Turbine Rotor—Speed reducer—DC generator—Voltage Invertor (Current invertor)—Supply Network".

The advantage of this scheme is that the energy of rotation of the free turbine shaft can be recovered into the supply network. However, the creation of a load torque on the GTU's FT shaft presupposes the flow of currents through the anchor winding of the generator and, accordingly, the power converter, proportional to the load torque in accordance with the loading program. In addition, this means that the energy of the mechanical movement of the FT shaft will be dissipated in the generator winding and the power elements of the power energy converter.

Taking into account that semiconductor power (thyristor and transistor) converters operate in the pulse mode, the main heat dissipation energy falls on the anchor winding of the DC generator. The voltage drop and heat dissipation power in semiconductor switches are generally neglected and efficiency is reduced.

A loading scheme is also possible due to a change in the three-phase active load at the inverter output; however, such a loading model has the same disadvantages as models with an active load resistor in the anchor circuit of the DC generator. Moreover, the load of the inverter must be three-phase symmetrical, which further complicates and increases the cost of the load unit.

### 2.1.3. Loading Model According to the System "Free Power Turbine Rotor—Speed Reducer—Synchronous Generator—Load Resistor"

This model of loading the GTU's FT shaft fully complies with the concept of loading during testing of combined-cycle gas turbine power plants. The scheme of the loading model is shown in Figure 3.

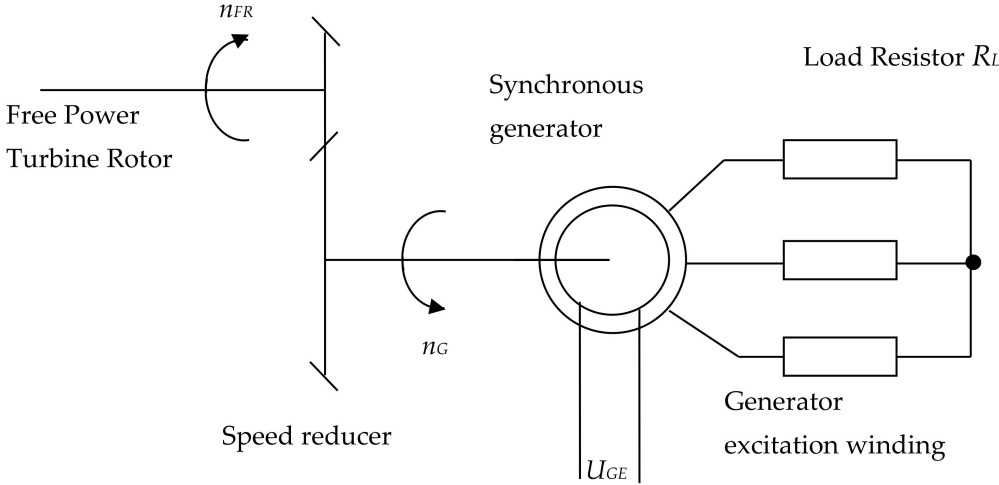

**Figure 3.** Scheme of the loading model "Free Power Turbine Rotor—Speed reducer—Synchronous generator—Load Resistor".

The output shaft of the speed reducer rotates at a speed $n_G$, not exceeding 314 rad/s (3000 rpm), with one pair of poles of a synchronous generator (*SG*).

In the three-phase stator winding of the *SG*, a three-phase voltage $U_{SG}$ is induced, the frequency of which is proportional to the rotation speed $n_G$, and the voltage is determined by the voltage value $U_{GE}$ of the excitation winding of the *SG*.

The *SG* is loaded by connecting a symmetrical active load in the form of star-connected resistors. The loading power will be determined by the size of the resistors connected using the switching devices:

$$N_{SG} = \frac{\sqrt{3}U_{SG}^2}{R_L},\tag{5}$$

where $R_L$—the resistance of one section of the phase load of the *SG* stator.

Taking into account the generally accepted assumptions [26], the load torque on the *SG* shaft is:

$$M_{SG} = 60 \cdot N_{SG}/2\pi\, n_{SG}.\tag{6}$$

### 2.1.4. Loading Model According to the System "Free Power Turbine Rotor—Speed Reducer—Synchronous Machine—Electric Power Drive—Supply Network"

Such a model of loading a gas turbine unit is structurally universal as applied to testing a gas turbine engine, both as a part of a gas pumping unit and a combined-cycle gas turbine power plant.

In the first case, the three-phase stator winding of a synchronous machine is connected to the electric power drive of comparable power, which is connected to an industrial supply network with a voltage of 6 or 10 kV (Figure 4).

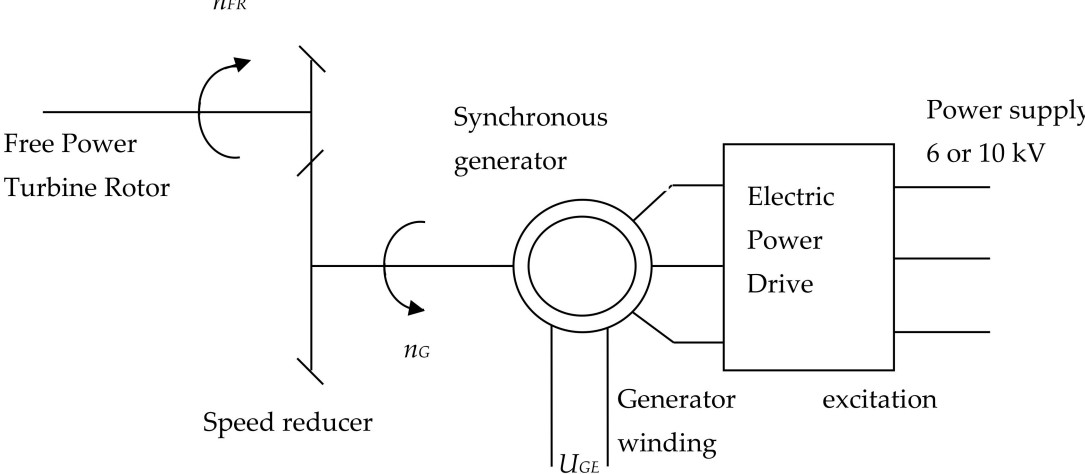

**Figure 4.** Scheme of the loading model "Free Power Turbine Rotor—Speed reducer—Synchronous machine—Electric Power Drive—Supply Network".

In the second case, the SG is a part of the combined-cycle gas turbine power plant, so it is enough to connect it to the electric power drive.

The disadvantages of the electromechanical loading method include the following:

1. The presence of three power modules (Speed reducer, synchronous machine, and electric power drive system), which significantly complicate the loading model and thereby reduce the reliability of the loading system functioning, complicating maintenance, repair, etc.;
2. Large dimensions and weight of the loading system, mainly due to a synchronous machine (about 80–140 tons), which requires the design and installation of a frame structure on a special foundation;
3. The presence of a speed reducer (even the simplest cylindrical single-stage reducer) leads to the appearance of a backlash and a gap in the kinematic chain, which reduces the accuracy of power transmission from the FT shaft to the synchronous machine shaft;
4. The model is not equipped with a loading automation system, including measuring, control, protection, and other functions provided for by the requirements for the modern loading system;
5. The high cost of the power elements of the loading system.

*2.2. Hydromechanical Loading Models*

Nowadays, a hydraulic loading system is the most common way of creating a load on the shaft of powerful tested power plants, since it has many advantages over the method of electromechanical loading. The undoubted advantages of hydraulic brake-dynamometers, providing hydraulic loading of gas turbines, include relatively small dimensions, weight, relatively low price and sufficiently high functionality, which allows not only to implement the necessary loading modes, but also to provide high-precision control of the main parameters of the loading system, such as the shaft's rotation speed and the torque of load on the shaft.

2.2.1. Loading Model According to the System "Free Power Turbine Rotor—Speed Reducer—Pump—Control Valve"

In such a loading scheme, a water pump or a set of them with parallel connection is used. Parallel connection increases the flow rate. A control valve is used as a control device, for example, with control from an electric drive. In the case of using several pumps in parallel, it is possible to vary either their connection, or to adjust the characteristics of the pipeline by changing the position of the regulating body of the actuator (valve). This

principle of regulation creates additional possibilities for expanding the functionality of the bench, but increases its dimensions and requires the transfer of mechanical movement to a group of pumps operating in parallel.

Note that the scheme in Figure 5 assumes the presence of a recycled water supply system (RWSS), which means that it takes some time to cool the water that has absorbed heat when overcoming the hydraulic resistance of the line.

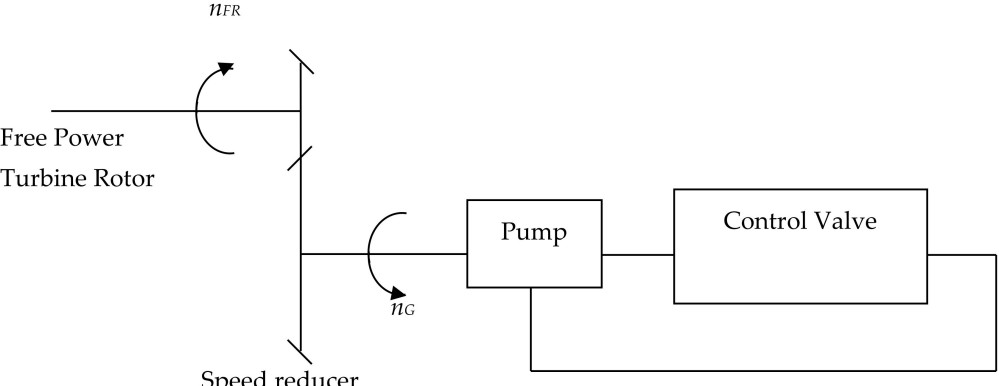

**Figure 5.** Scheme of the loading model "Free Power Turbine Rotor—Speed reducer—Pump—Control Valve".

### 2.2.2. Loading Model According to the System "Free Power Turbine Rotor–Hydraulic Brake"

This loading model of the gas turbine unit is dominant today. It, like all models, has its disadvantages associated with the practical (although it is controversial) impossibility of utilizing the energy of converting the mechanical energy of rotation of the GTU's FT shaft into thermal energy of the heated fluid at the outlet of the hydraulic brake. A generalized scheme of this loading model is shown in Figure 6.

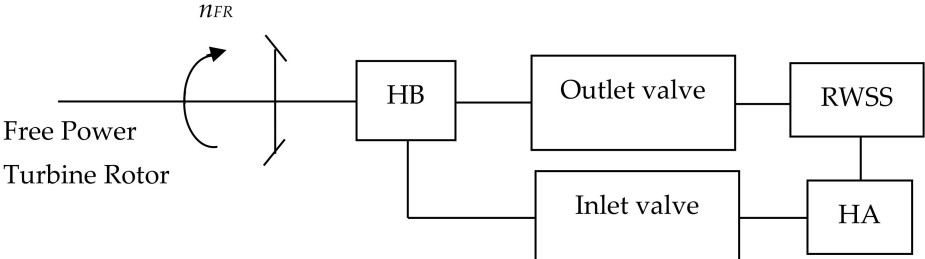

**Figure 6.** Scheme of the loading model "Free Power Turbine Rotor–Hydraulic Brake".

Note that there is no speed reducer in the energy conversion technological chain. This allows saving on capital investments and operating costs as well as on the occupied areas. In addition, this loading model removes such problems as accounting for the speed reducer efficiency (as a function of the rotation speed), estimating the elastic-viscous vibrations of the kinematic transmission (with analyzing the stability of gas-dynamic characteristics of gas turbines), etc.

In order to create an input pressure into the hydraulic brake (HB) of the working fluid (water purified from impurities and gases), such a model requires the creation of a recycled water supply system, which includes a hydroaccumulator (HA).

Controlling the loading of the FT shaft in such a loading model assumes the possibility of varying:

- the pressure of the fluid at the outlet of the HA;
- the throughput (geometry of the water line) at the inlet and outlet of the HA using adjustable inlet and outlet valves (according to Figure 6).

The principle of operation of the hydraulic brake knowingly establishes a specific dependence between the degree of opening of the inlet valve and the degree of closing (opening) of the outlet valve. This means that in practice only two control variables can be varied. However, even this is quite enough to realize the required "Pressure—Consumption" characteristic of the hydraulic brake at each moment of the time of the GTU's FT shaft loading program.

Thus, the presented GTU loading model is very attractive from all sides—a minimum of elements involved in the technological loading process, a generalized minimum of weight and size indicators, a cost minimum, sufficient functionality, and controllability.

Further, we will take into account the "Free Power Turbine Rotor—Hydraulic Brake" model as the basic for the stand loading systems.

However, the disadvantages of this loading method include the following:

1. The need for the design and manufacture of a recycled water supply system, which includes a fairly long pipeline with a lot of elements of shut-off and control valves, a hydraulic accumulator, a pressure tank, and other elements;
2. Limited possibilities of formation of GTU's FT loading programs, which are also in a rather narrow range of variation of the shaft rotation frequency of FT;
3. Inertia of control of electro-hydraulic valves at the inlet and outlet of the hydraulic brake, which does not allow for a quick change of the loading mode;
4. Impossibility or inexpediency of utilizing the energy of loading the FT shaft, since during the test cycle the mechanical energy of the shaft is converted into thermal energy of heating the water passing through the hydraulic brake by about 30 °C.

Nevertheless, there are investigations where the mathematical part that represents the behavior of the centrifugal pump in the form of designed software combines with the hardware part, which imitates hydraulic loads according to the control program [27].

At the same time, rather strict requirements are imposed on the bench's loading system. The loading system device must:

- provide the range and character of load changes in accordance with the test program, creating a braking torque in stationary and dynamic modes of GTE testing;
- ensure stable operation in all modes from the start of the gas turbine engine to the maximum load mode, corresponding to the field of the load characteristic for each of the operation options (with maintaining a constant shaft speed ($n = const$); maintaining a constant loading power ($N = const$); according to the dependence of power on the rotor speed, close to cubic ($N = f\{n\}$));
- ensure the maintenance of a constant value of the consumed power in the stationary test mode of the gas turbine engine;
- allow the start of the GTE in any idle mode of the GTE in the entire interval for an unlimited time (within the resource) with preliminary filling of the loading system with water and/or without preliminary filling;
- have a system for measuring the frequency of rotation of the shaft, providing the required measurement error;
- be equipped with a torque measuring system using a force-measuring transducer;
- be equipped with a device for calibrating the torque measuring system;
- provide for the possibility of its reconciliation (alignment) with respect to the GTE.

E.g., Table 1 shows the main parameters of one of the gas turbine engines planned for testing at the bench with using a hydromechanical loading system.

**Table 1.** GTE parameters.

| Tested Mode | Rotation Speed of the Turbine Shaft, rpm | Turbine's Shaft Power, kW |
| --- | --- | --- |
| Minimum (with input temperature +15 °C) | 4400–4600 | 3790 |
| Nominal (with input temperature +15 °C) | 6400–6600 | 11,760–12,240 |
| Maximum (with input temperature +5 °C) | 6630–6830 | 13,068–13,596 |

## 3. Monitoring, Control, and Protection System of the Loading Device

The loading device has to provide:

- control of the hydraulic brakes parameters and its auxiliary systems;
- automatic regulation of the volume of the air cushion of the HA and the value of the water pressure at the hydraulic brake's inlet.

To ensure the test modes for the GTU, it is necessary to:

- select the required operating mode of the loading system controller;
- periodically transmit to the loading system controller the values of the desired load value (rotation speed or torque).

The bench's control system is divided into an upper and a lower level.

At the lower level, there is a primary processing of information coming from the actuators and primary converters, as well as the development of control actions. The lower level allows to control the bench subsystems, as well as collect and initial processing of information from the primary converters of the bench. Information from the lower layer goes to the upper and vice versa, via the Ethernet network using the TCP/IP protocols.

The upper level is intended:

- to form and make decisions at the system's level at a whole;
- to issue control commands to the test bench using an intuitive interface based on mnemonic diagrams on the monitor screen;
- to display the status of all actuators and measured values;
- to record all events, with the ability to graphically view received data;
- to mathematical process results with the subsequent issuance of protocols and graphs.

As a result of studying the control object and the technological process of testing, taking into account the required speed of the system as well as the number of measured parameters and controlled elements of the bench and GTE, it was decided to build the lower level based on the PXI (PCI eXtensions for Instrumentation) [28] and SCXI [29] standards.

The lower level consists of the system of registration of slowly changing parameters (SCP-system) and the system of registration of rapidly changing parameters (RCP-system).

The SCP-system is designed to measure and record signals from sensors of temperatures, pressures, rotation frequency, flow rates, and efforts as well as the position of the actuating elements of the bench. The parameters are grouped into two subsystems: analog data input and discrete data input. The SCP-system allows to issue about 200 control signals.

The RCP-system is designed to measure and record the signals from vibration sensors, pulsations and deformations sensors. Unlike the SCP-system, the RCP-system provides the higher frequency interrogation of transducers, with a wider dynamic and frequency range as well as area of operation of the input modules. The system is based on the dynamic input modules PXI NI 4472 standard. The RCP-system allows registering up to 24 dynamically changing signals.

The bench control system uses primary converters to obtain information about the state of the bench parameters.

Measuring channels perform a complete function from the perception of the measured value to obtain the result of its measurements, expressed by a number or the corresponding code. Moreover, they perform to receive an analog signal, one of the parameters of which is a function of the measured value

The measuring channel can be generalized as shown in Figure 7.

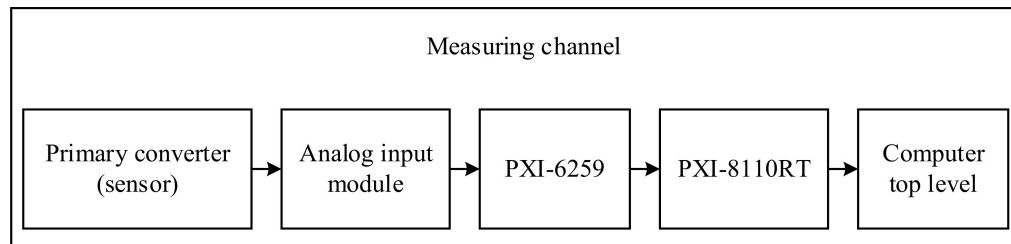

**Figure 7.** Generalized structure of the measuring channel.

The registering of the parameters and GTU control system's signals is carried out in a continuous mode of all the test phases from each start to stop of the GTE.

## 4. The Task of the Bench Hardware-in-the-Loop Simulation Module Software Developing

### 4.1. Formulation of the Problem

Strict requirements for the quality indicators of the generated energy necessitate the introduction of significant and fundamental modifications to the existing prototypes of gas turbine units. However, at present, when designing and debugging automatic control systems of power turbines, the electric power system as a whole is taken into account very approximately [6,8].

The development of modeling benches for power facilities, which allow for solving various functional tasks, is relevant [30]. To take into account the main modes of operation, the test bench is connected to a computer model of the power system.

During the test, the bench control system must calculate the value of the rotational speed that must be maintained in the mode and transmit this value to the dynamometer controller.

The torque in the modes is not regulated. Its values depend on the characteristics of the tested engine and atmospheric pressure.

So, Figure 8 shows the dependence of the torque of a gas turbine engine (Table 1) on the rotation speed at start-up.

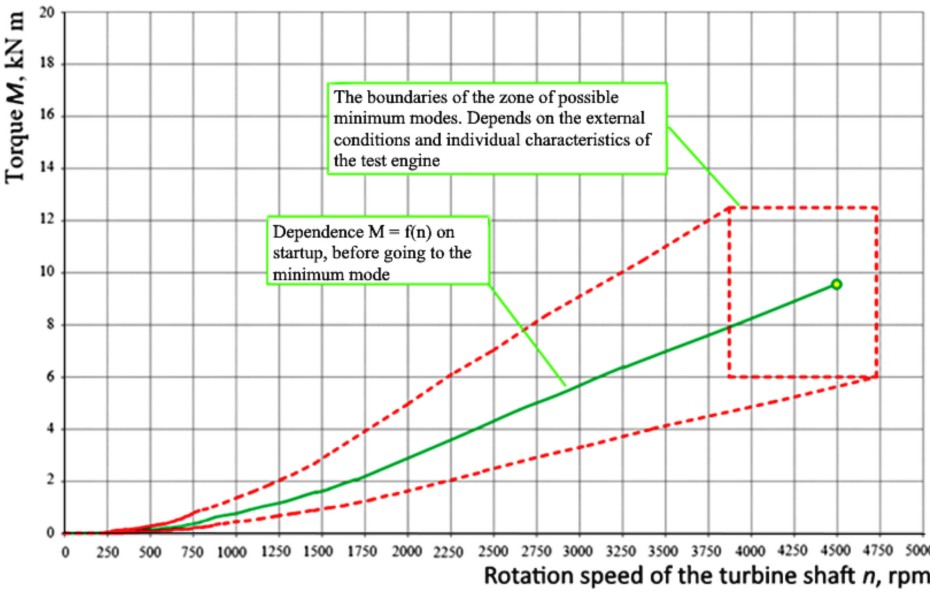

**Figure 8.** Dependence of the GTE torque on the rotation speed at start-up.

The cyclograms of changes in the rotation speed and torque versus time, the dependence of the torque on the rotation speed during tests of this type of engine, are shown in Figures 9–11.

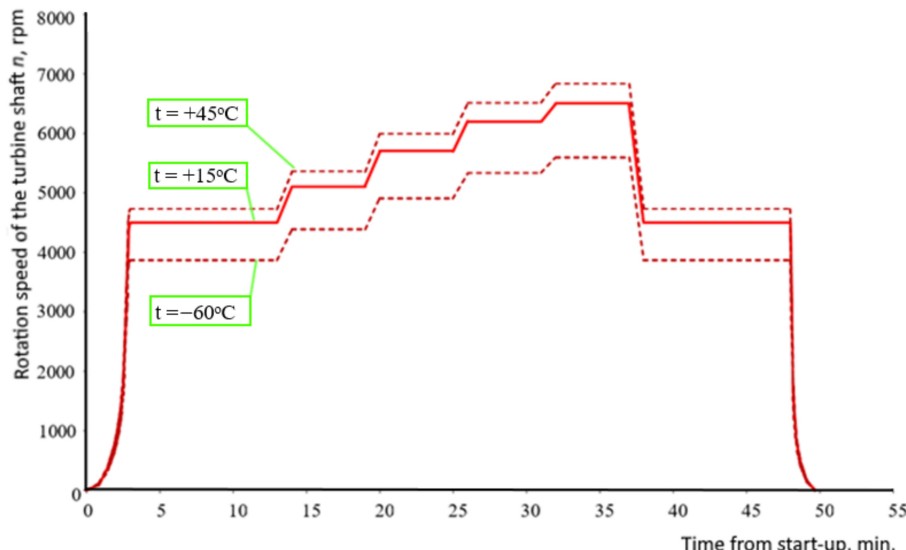

**Figure 9.** Cyclogram of GTE testing with dependence of rotation speed on time.

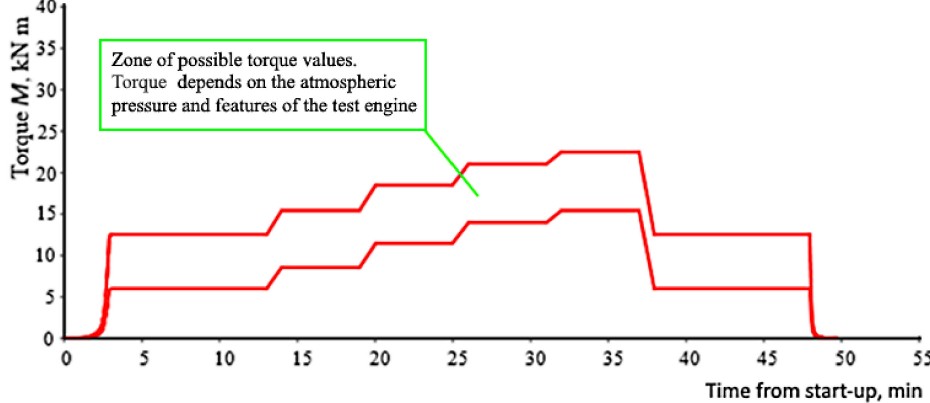

**Figure 10.** Cyclogram of GTE testing with dependence of torque on time.

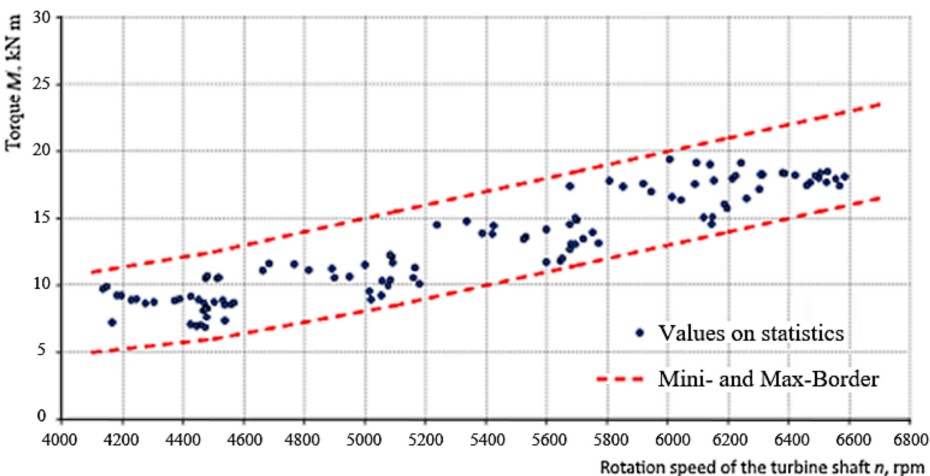

**Figure 11.** Range of torque values depending on the GTE rotation speed in test modes.

The parameters of the operating modes of the specified GTE during testing are given in Table 2.

**Table 2.** The parameters of the operating modes of the specified GTE.

| Mode | Operating Mode's Time, min. | Rotation Speed, Given to Temperature +15 °C, rpm | Rotation Speed Under Extreme Conditions, Rpm (at Temperature −60 °C) | Rotation Speed Under Extreme Conditions, Rpm (at Temperature +45 °C) | Torque, kN·m (min.) | Torque, kN·m (max.) |
|---|---|---|---|---|---|---|
| Start-up | 3 | - | - | - | - | - |
| 1st minimum mode | 5 | $4500 \pm 100$ | $3870 - 100$ | $4728 + 100$ | 6.0 | 12.5 |
| 2nd minimum mode | 5 | $4500 \pm 100$ | $3870 - 100$ | $4728 + 100$ | 6.0 | 12.5 |
| Mode 10,000 | 5 | $5100 \pm 100$ | $4386 - 100$ | $5359 + 100$ | 8.5 | 15.5 |
| Mode 10,250 | 5 | $5700 \pm 100$ | $4902 - 100$ | $5989 + 100$ | 11.5 | 18.5 |
| Mode 10,500 | 5 | $6200 \pm 100$ | $5332 - 100$ | $6515 + 100$ | 14.0 | 21.0 |
| Nominal mode 12 MW | 5 | $6500 \pm 100$ | $5590 - 100$ | $6830 + 100$ | 15.5 | 22.5 |
| Maximum mode 13 MW | 5 | $5630 \pm 100$ | $5788 - 100$ | $7072 + 100$ | 16.5 | 23.5 |
| 3rd minimum mode | 5 | $4500 \pm 100$ | $3870 - 100$ | $4728 + 100$ | 6.0 | 12.5 |
| Stop | - | - | - | - | - | - |

The test cyclograms show that the boundaries of the zone of possible minimum GTE modes depend on external conditions and individual characteristics of the engine being tested.

Taking into account the operating factors in specific technological conditions (changes in the configuration and topology of the power system, changes in the operating modes of the supply network, the load's specifics, etc.) at the stage of testing these objects will allow the operator to understand the technical and technological limitations at the stage of the operation. On the other hand, it will be opportunity to formulate requirements for equipment operation modes, maintain a database of parameters of the power system's elements from the test stages, and formulate requirements for equipment maintenance and repair measures.

Thus, it becomes possible to combine the stages of mathematical modeling of the power system and Hardware-in-the-Loop (HIL) tests (the stage of testing models and prototypes) [30].

From a software and technical implementation point of view, the test complex includes five applications shown in Figure 12.

To obtain the required mathematical model of the power plant, first of all, the selection of the parameters required for its identification on the basis of experimental data is carried out. Then, the mathematical model is identified and checked for adequacy according to Theil's criterion [31]. The obtained parameters of the model are written to a file of the format *.xml* [32].

The results of the identification of the power plant are the values of the coefficients of the differential equations of the model and the static characteristics.

### 4.2. Energy Source Models

Let us consider mathematical models of gas turbine units and gas pumping units operated at enterprises. To study power plants, it is advisable to use simplified identification models [8,32], which, nevertheless, should adequately reproduce transient processes in power plants and nonlinearity of characteristics when changing the operating mode of the plants. Power plants interact with synchronous generators mechanically, thus, bringing the models of power plants to a single generalized form, as shown in [8], is not required.

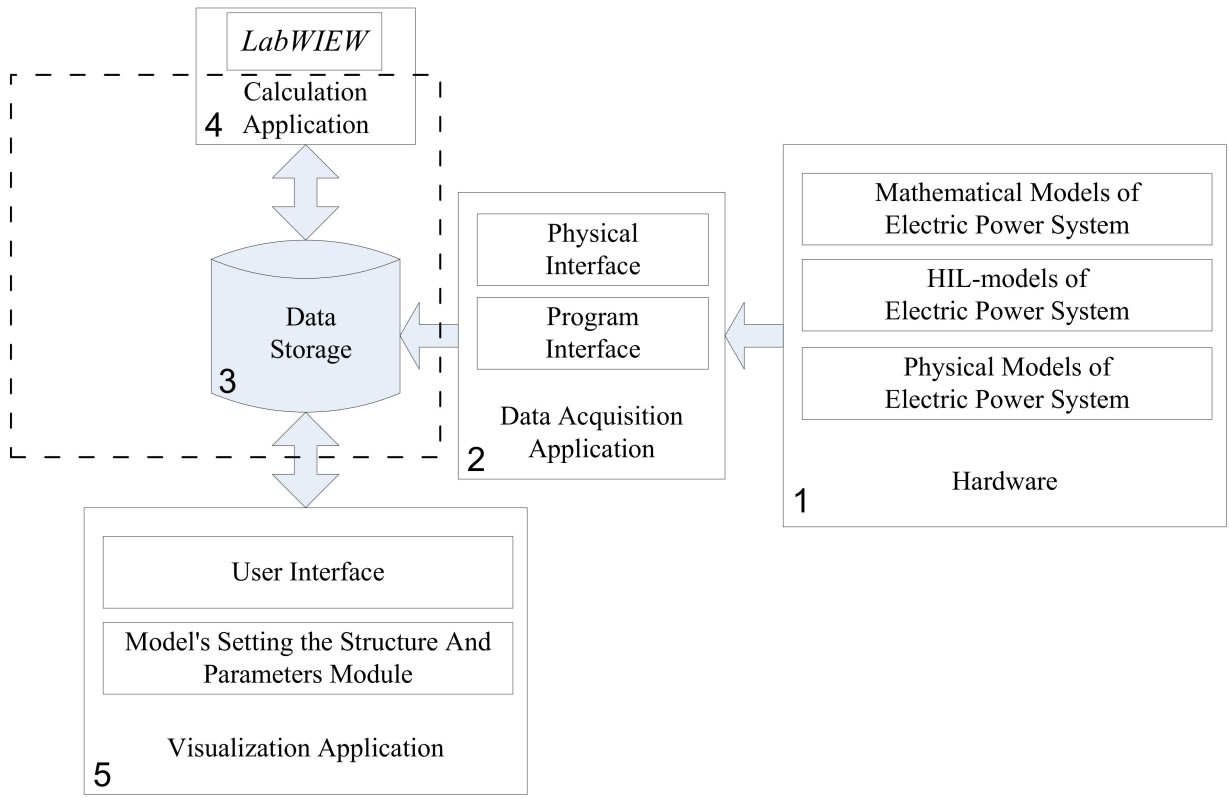

**Figure 12.** Software applications of the test computer complex.

4.2.1. Nonlinear Model of a Gas Turbine Unit

The model takes into account the accumulation of energy in the rotating masses of the rotors of a two-shaft gas turbine unit.

$$
\begin{cases}
\dot{A}_{DI} = \frac{(A_{DIZ} - A_{DI})}{T_{DI}}, \\
G_{FS} = f(A_{DI}), \\
\dot{G}_F = \frac{(G_{FS} - G_F)}{T_{GF}}, \\
n_{TS} = f(G_F), \\
\dot{n}_T = \frac{(n_{TS} - n_T)}{T_{NT}} \\
N_E = f(n_T), \\
\Delta \dot{N}_C = \frac{K_N \frac{\dot{N}_E - N_G}{n_{FR}} - \Delta N_C}{T_N}, \\
\dot{n}_{FR} = \frac{N_E - N_G}{T_{NFR} \cdot n_{FR}} + \Delta N_C,
\end{cases}
\tag{7}
$$

where $A_{DI}$—gas regulator rotation angle; $A_{DIZ}$—specified gas regulator rotation angle; $G_F$—fuel consumption; $G_{FS}$—fuel consumption based on the static characteristic; $T_{DI}$—gas regulator time constant; $T_{GF}$—time constant of fuel consumption; $T_{NT}$—turbocompressor rotor time constant; $T_N$—time constant of the influence of the rate of change in the power imbalance on the rotation speed of the free turbine; $T_{NFR}$—free power turbine rotor time constant; $n_T$—turbocompressor rotor speed; $n_{FR}$—free power turbine rotor speed; $n_{TS}$—turbocompressor rotor speed based on the static characteristic; $N_E$—available power of free power turbine; $N_G$—power consumption of free power turbine; $\Delta N_C$—value reflecting the influence of the derivative of the power imbalance on the free power turbine rotor; $K_N$—coefficient of amplification of the rate of change of power imbalance.

4.2.2. Nonlinear Model of the Gas Pumping Unit

The model is designed to organize the procedure for tuning control systems of a gas pumping unit. This model takes into account the accumulation of energy in the rotating masses of the rotors of a gas pumping unit of a two-shaft design [6].

$$
\begin{cases}
\dot{A}_{DI} = \frac{(A_{DIZ} - A_{DI})}{T_{DI}}, \\
G_{FS} = f(\dot{A}_{DI}), \\
\dot{G}_F = \frac{(G_{FS} - G_F)}{T_{GF}}, \\
n_{TS} = f(\dot{G}_F), \\
\dot{n}_T = \frac{(n_{TS} - n_T)}{T_{NT}}, \\
N_E = f(n_T), \\
n_{FRZ} = f(N_E), \\
\dot{n}_{FR} = \frac{(n_{FRZ} - n_{FR})}{T_{NFR} \cdot n_{FR}}.
\end{cases}
\tag{8}
$$

where $n_{FRZ}$—free power turbine rotor speed based on the static characteristic.

*4.3. The Turbine Units Modelling*

The parameters of the models of power plants are obtained using the Calculation Application (Block 4 in Figure 12) [32] and are entered into the simulation module on the LabVIEW platform through files of the *.csv* and *.xml* formats. Further, the software and hardware modeling of the electric power system is carried out. Moreover, the output of the static and dynamic characteristics of the main parameters of power plants is obtained.

Simulation settings include:

1. the step of saving the experiment data—a step in time through which the parameters of the elements are removed, saved and displayed in the form of graphs in the window of simulation results. This parameter affects the accuracy of displaying the graphs of gas pumping unit parameters;
2. simulation time—total simulation time;
3. the step of solving differential equations—a step in time with which the solution of differential equations occurs. This parameter affects the accuracy of the calculation of transients.

Input data for the Calculation Application Block includes:

Parameters model power plant (file of format *.xml* with the values of the coefficients and constants set the time, and static characteristics);

Parameters control system (file of format *.xml* with values control system coefficients and static characteristics);

Load's model (file of format *.csv* with parameters of the electric power system's elements).

This module is based on the concept of modular programming: a large task was divided into a number of simple sub-tasks, after which virtual instruments (VIs) were created to perform each of these sub-tasks, and then combined on a block diagram of a higher-level device that performs the application the task as a whole.

The structure of the *gtuSimulatorProject.lvproj* project [32] is shown in Figure 13.

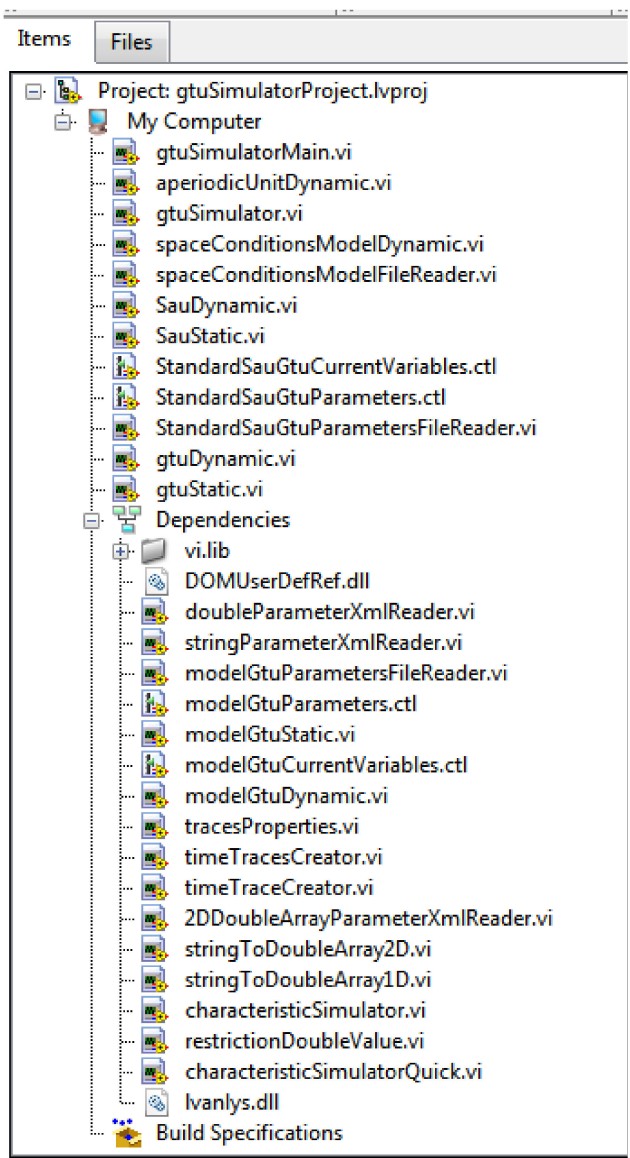

**Figure 13.** The structure of the *gtuSimulatorProject.lvproj* project.

The *GtuSimulatorMain.vi* block diagram is shown in Figure 14.
The VIs and subVIs of *GtuSimulatorMain.vi* provide:

- reading the parameters of the power system from the *.csv* file and extraction of the matrix of model coefficients in the state space, the state vector of the initial steady state, and the state vector of the beginning of the transient process;
- calculating the static mode of the GTU model and the control system of the GTU (with input data of $N_G$, $n_{FRZ}$, GTU model parameters, and parameters of the control system of the GTU);
- calculating the dynamic mode of the GTU and of one iteration of an arbitrary calculation step (with input data of the control system of the GTU variables, GTU model parameters, GTU variables, load power in dynamics, and the step of solving differential equations);
- carrying out the interpolation according to static characteristics, which allows to determine the main parameters of the gas turbine unit (using one-dimensional polynomial Hermitian interpolation built-in LabVIEW).

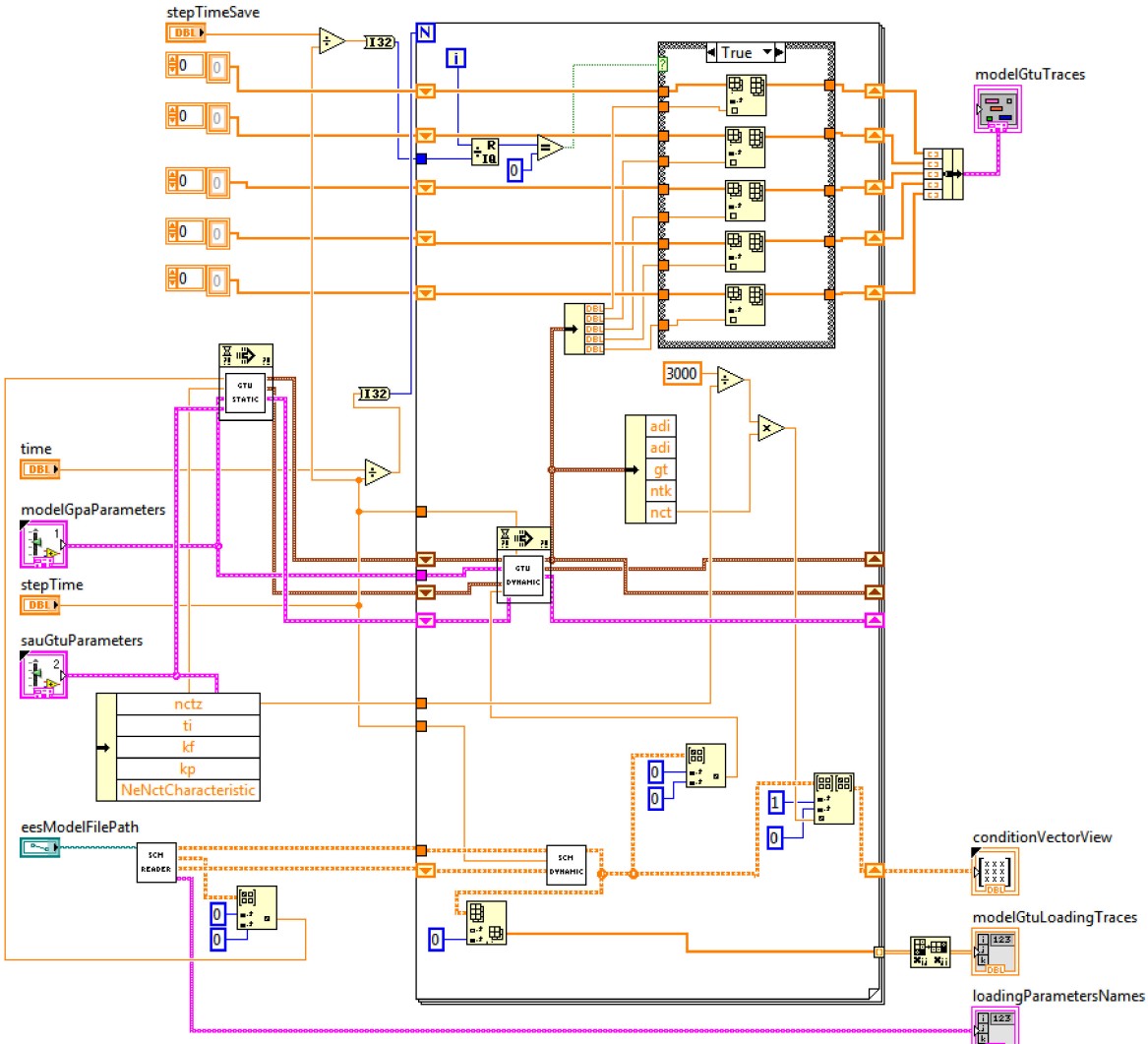

**Figure 14.** Block diagram of the virtual instrument *gtuSimulator.vi*.

The system monitors that the simulation time should not be less than the step of saving the experiment data, and the step of saving should not be less than the step of solving differential equations.

Mathematical modeling involves 2 stages:

(1) calculation of the static mode;
(2) calculation of the dynamic mode.

Parameters such as power consumption, gas regulator rotation angle, and free power turbine rotor speed ensure interaction between mathematical models of a gas turbine unit, an automatic control system of a gas turbine unit, and a power system.

## 5. Results

Let us consider the comparison of experimental and model data for a nonlinear model of a gas-pumping unit.

### 5.1. Step Increase Load

The study was carried out by feeding the experimental values of gas regulator rotation angle and the load power to the input of the GPU model. The simulation time was 79.9 s, and the step of saving the experiment data was 0.1 s.

Figures 15–17 show the dependences of the turbocompressor rotor speed, free power turbine rotor speed, and the fuel consumption of GPU on the time of the transient process, obtained experimentally and during simulation.

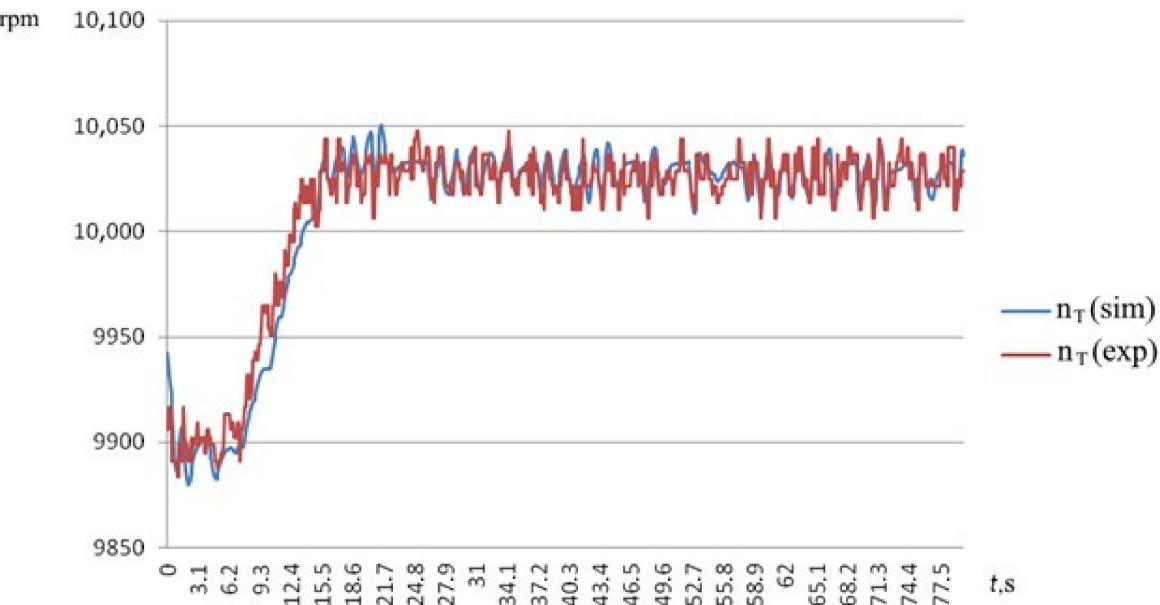

**Figure 15.** Dependence of the turbocompressor rotor speed on the time of the transient process during the step increase load (red line—experimental, blue line—model).

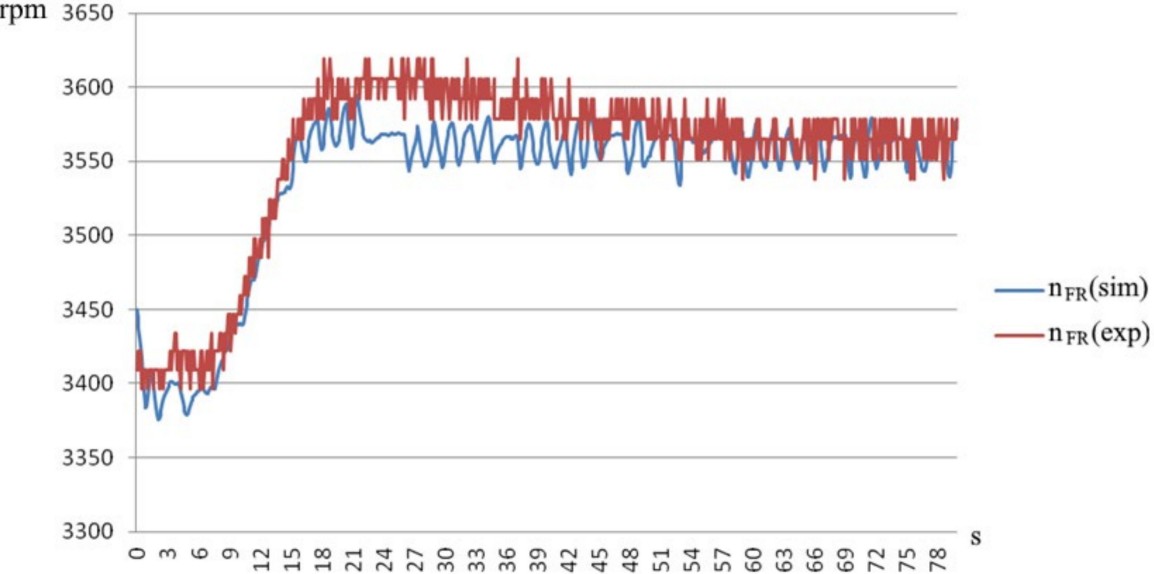

**Figure 16.** Dependence of the free power turbine rotor speed on the time of the transient process during the step increase load (red line—experimental, blue line—model).

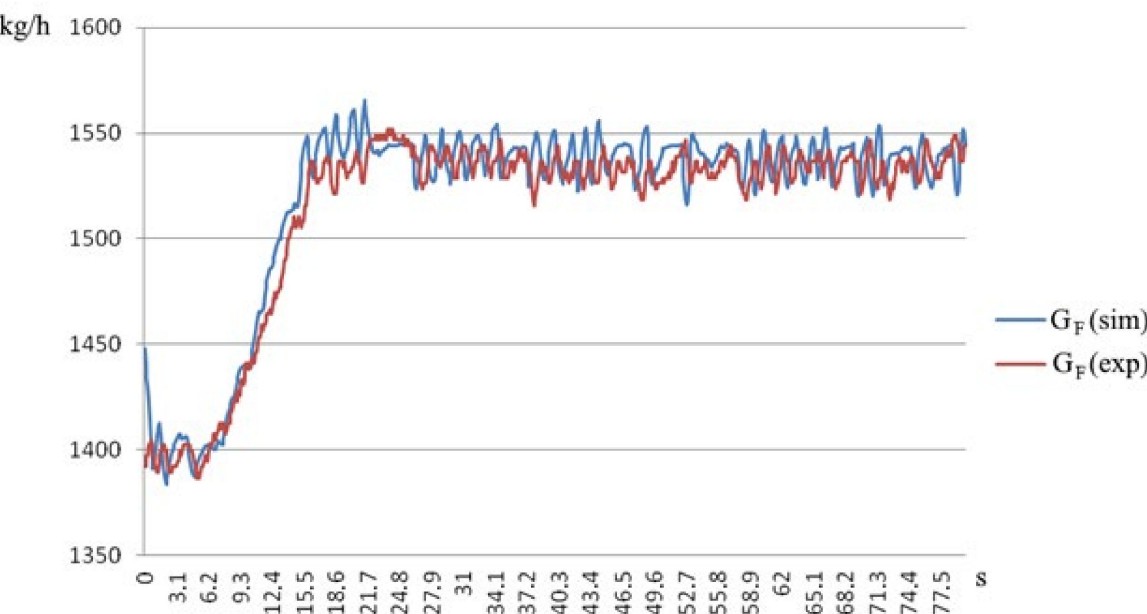

**Figure 17.** Dependence of the fuel consumption on the time of the transient process during the step increase load (red line—experimental, blue line—model).

### 5.2. Step Decrease Load

The study was carried out by feeding experimental values of the gas regulator rotation angle and the load power to the input of the GPU model. The simulation time was 299.9 s, and the step of saving the experiment data was 0.1 s.

Figures 18–20 show the dependences of the turbocompressor rotor speed, free power turbine rotor speed, and the fuel consumption of GPU on the time of the transient process, obtained experimentally and during simulation.

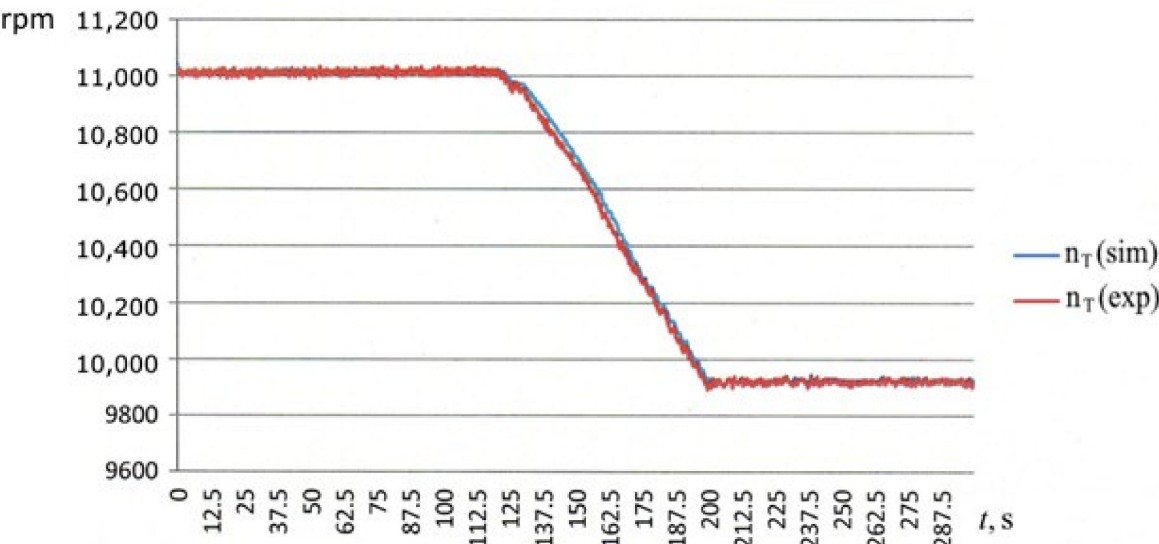

**Figure 18.** Dependence of the turbocompressor rotor speed on the time of the transient process during the step decrease load (red line—experimental, blue line—model).

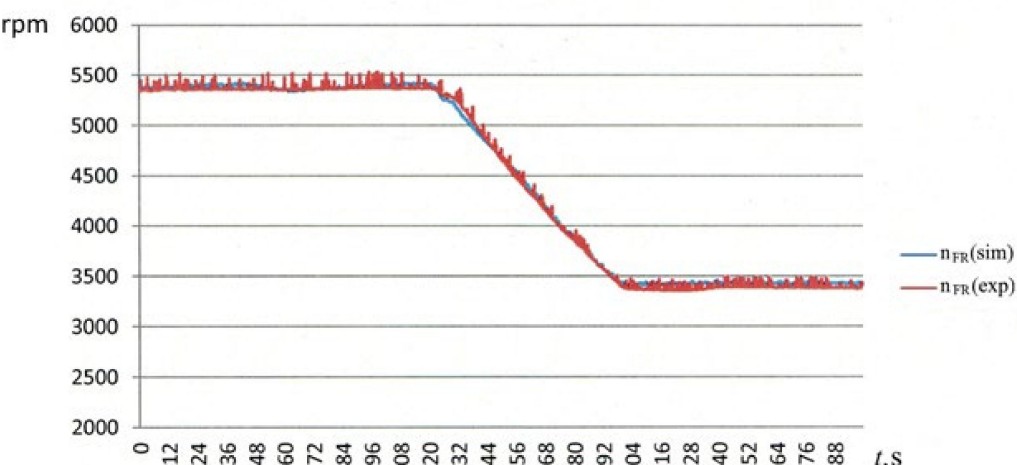

**Figure 19.** Dependence of the free power turbine rotor speed on the time of the transient process during the step decrease load (red line—experimental, blue line—model).

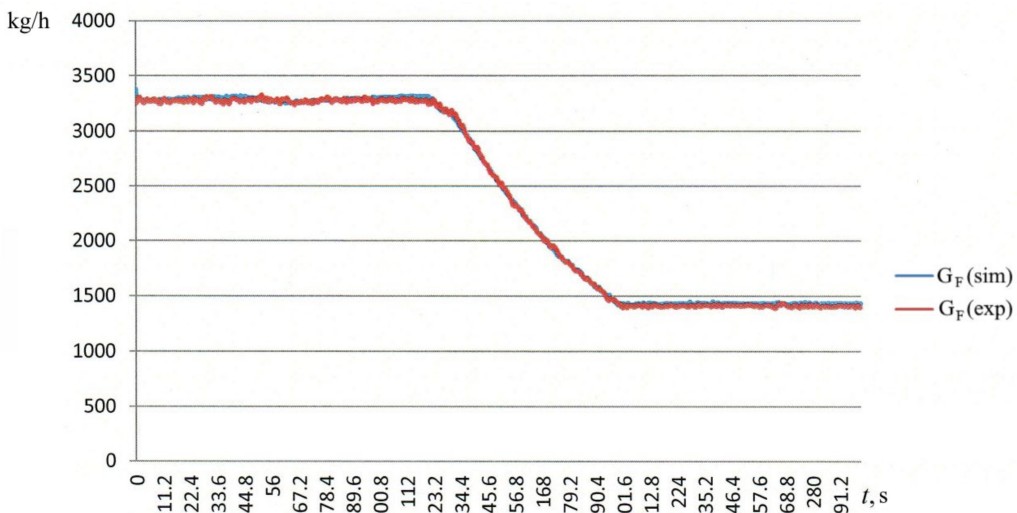

**Figure 20.** Dependence of the fuel consumption on the time of the transient process during the step decrease load (red line—experimental, blue line—model).

To determine the quality and accuracy of the designed mathematical models, the Theil uncertainty coefficient $C_T$ is used, which measures the degree of mismatch between the values of the generalized process parameter, determined experimentally, with the values of the generalized process parameter, determined as a result of the calculation using the model [31,32]:

$$C_T = \frac{\sqrt{\frac{1}{n}\sum_{i=1}^{n}\left(x_i^{(\text{exp})} - x_i^{(\text{sim})}\right)^2}}{\sqrt{\frac{1}{n}\sum_{i=1}^{n}\left(x_i^{(\text{exp})}\right)^2} + \sqrt{\frac{1}{n}\sum_{i=1}^{n}\left(x_i^{(\text{sim})}\right)^2}}, \tag{9}$$

where $x_i^{(\text{exp})}$—experimental value, $x_i^{(\text{sim})}$—the value calculated by the functional model, and $n$—number of experimental values that were used to synthesize the model (the number of experimental values must coincide with the number of values calculated by the functional model).

Table 3 shows the obtained Theil coefficients for the turbocompressor rotor speed $C_T(n_T)$, for the free power turbine rotor speed $C_T(n_{FR})$ and for the fuel consumption $C_T(G_F)$ of the investigated models of power plants.

**Table 3.** Values of Theil uncertainty coefficient.

| Tested Model | $C_T(n_T)$% | $C_T(n_{FR})$% | $C_T(G_F)$% |
|---|---|---|---|
| Nonlinear gas pumping unit model (step increase load) | 0.05 | 0.3 | 0.4 |
| Nonlinear gas pumping unit model (step decrease load) | 0.04 | 0.5 | 0.4 |

Based on the results of calculating the values of the Theil uncertainty coefficient for the main compared parameters, it can be concluded that the constructed mathematical models of power plants are adequate. Most of the Theil coefficient values do not exceed 2–3%; therefore, these parameters, determined experimentally and calculated by the model, have good convergence [31].

## 6. Discussion

The object of the study is the principles and models of loading gas pumping units based on various methods of converting the mechanical energy of rotation of the shaft of a free turbine into a different type of energy, providing the required loading modes in accordance with the specified programs and satisfying the set of specified constraints.

As a result of the study, a review of existing technical solutions was carried out; a scientific analysis of the possibilities of implementing loading devices based on hydraulic, pneumatic, and electromechanical energy was given; and mathematical models of loading processes were presented.

Based on the results of the analysis of existing loading methods based on hydraulic, pneumatic, and electromechanical energy, the expediency of using the loading model "Free Power Turbine Rotor—Hydraulic Brake" as a load simulation is shown.

Recommendations for the creation of an automation system for the load testing of power plants have been developed.

Mathematical models and Hardware-in-the-Loop simulation models of power plants have been developed and tested. One of the most important factors that predetermine the effectiveness of the loading principle is the possibility that software implementation of the loading means using software control systems that provide the specified loading parameters of the gas turbine.

The main expected design and technical and operational indicators are:

- high functionality;
- versatility of the designed GTU's test automation system;
- high accuracy of measurement of such GTU parameters as the rotation speed of the free power turbine shaft, torque and power during the formation of a programmable load on the FT shaft.

The architecture of the simulation module allows for integration with other benches and test systems (Figure 21).

Test data of power plants can be stored in a production system and be made available for the manufacturers of power plants as well as for enterprises and customers, thereby providing integrated logistics support for power plants at all stages of their life cycle.

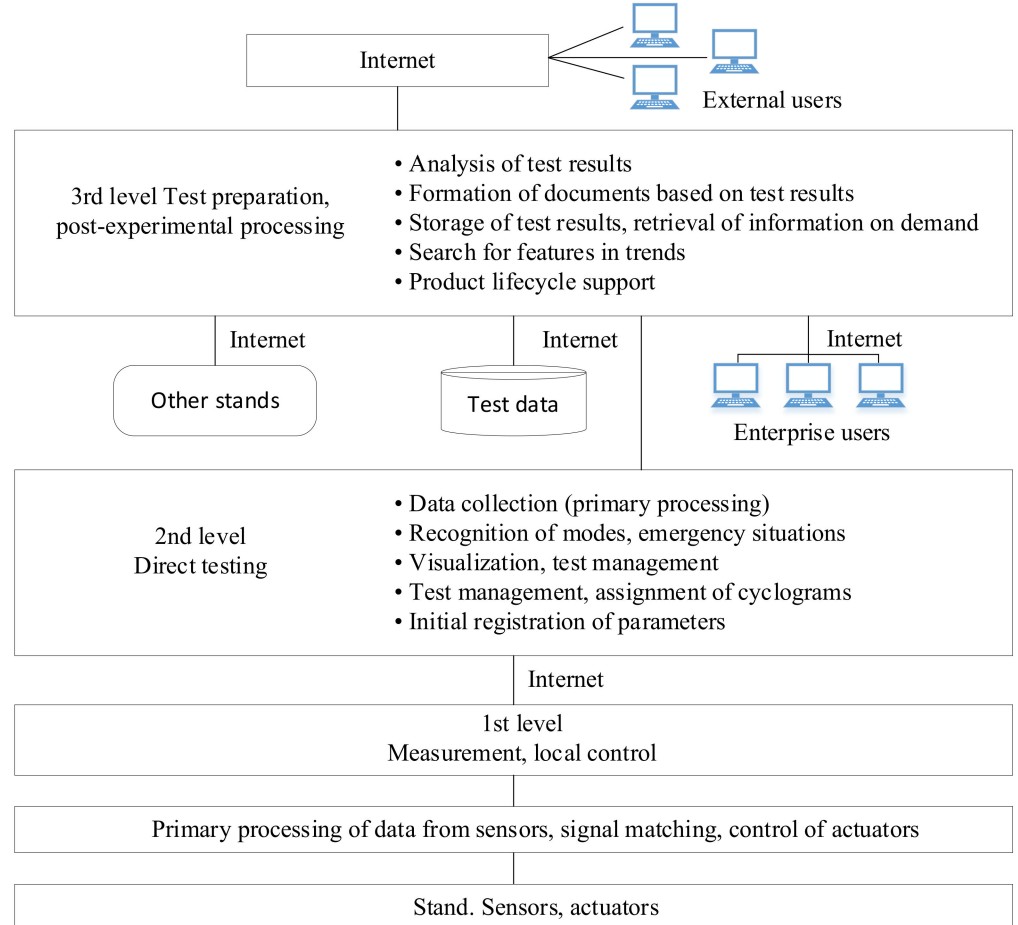

**Figure 21.** Structural and functional diagram of interactions of tests and test benches.

## 7. Conclusions

Qualitative multifactorial testing of power plants will improve energy efficiency and reduce environmental impact.

To solve the problem, along with a software and hardware complex that provides automation of the test process, it is necessary to single out a system that provides load regulation. The most common type of loading device for testing GTU is a hydraulic brake, which does not require a large range of regulation in terms of effective power and revolutions, but has physical limitations associated with cavitation.

Simulation of the load is also a research task associated with the construction of both mathematical and physical models. On the one hand, the advantages of physical models are:

- real course of physical processes in elements;
- the model is built on real objects, the behavior of which can be used as a standard for the verification of mathematical models.

However, the equipment often installed in laboratories has significantly lower power and lower rated voltage than equipment installed on real objects. When using the approach based on the similarity theory, in the initial data processing similarity coefficients are determined, and through them a transition is made from the results obtained at laboratory benches to the parameters of the operating modes of real objects [33]. The disadvantage of this approach in modeling the load system is that it does not take into account the change in the energy characteristics of the gas turbine engines in the non-nominal operating modes. The use of the HIL approach provides the development and testing of complex embedded real-time systems: the mathematical model of the control object works in real time, and the

control unit under test is connected to the hardware module at the bench and controls the object model.

**Author Contributions:** Conceptualization, A.P. and V.K.; methodology, A.P. and V.K.; software, A.R. and A.S.; validation, S.B. and Y.G.; formal analysis, S.B., Y.G., R.S. and P.B.; writing—original draft preparation, A.P.; writing—review and editing, A.P. and A.R.; visualization, A.R., S.B. and A.S.; supervision, A.P. and A.R. All authors have read and agreed to the published version of the manuscript.

**Funding:** This research was carried out with the financial support of the Ministry of Science and Higher Education of the Russian Federation in the framework of the program of activities of the Perm Scientific and Educational Center "Rational Subsoil Use".

**Institutional Review Board Statement:** Not applicable.

**Informed Consent Statement:** Not applicable.

**Conflicts of Interest:** The authors declare no conflict of interest.

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
