# Peer review of "Principles of Imitation for the Loading of the Test Bench for Gas Turbines of Gas Pumping Units, Adequate to Real Conditions"

_sustainability, doi:10.3390/su132413678_

Round 1

Reviewer 1 Report

1) Please try to improve the grammar and text quality. For instance instead of "ground test" -> "field test",
 "key mode" -> "pulse mode", etc.
2) Please try to cite more up-to-date references instead of old ones in the introduction part. For example you can add these two papers related to HIL methods:
L. Gevorkov, V. Vodovozov, T. Lehtla and Z. Raud, "Hardware-in-the-loop simulator of a flow control system for centrifugal pumps," 
2016 10th International Conference on Compatibility, Power Electronics and Power Engineering (CPE-POWERENG), 2016, pp. 472-477 and the second one L. Gevorkov, V. Vodovozov, T. Lehtla and Z. Raud, 
"PLC-based hardware-in-the-loop simulator of a centrifugal pump," 2015 IEEE 5th International Conference on Power Engineering, 
Energy and Electrical Drives (POWERENG), 2015, pp. 491-496
3) Please pay attention to diagram's symbols and designations. For instance in Fig. 1 try to add UG and RL.
4) Please rename Rsc to Rs in equation (2) so it becomes clear for the reader that you are describing equivalent anchor resistance.
5) Fig. 8, 9, 10, 11 are blurry so please try to improve the quality of the figures.
6) The paper represents an interesting approach and contains scientific novelty. Good job!

Author Response

1) Thank you very much for your comments. We fixed these mistakes.

2) We added the new links about HIL-simulations methods regarding close investigations. New sentences were added:

Nevertheless, there are investigations where mathematical part that represents behavior of centrifugal pump in the form of designed software combines with hardware part, which imitates hydraulic loads according to the control program [27].

[27] Gevorkov, L.; Vodovozov, V.; Lehtla, T.; Raud, Z. Hardware-in-the-loop simulator of a flow control system for centrifugal pumps. Proceedings of  2016 10th International Conference on Compatibility, Power Electronics and Power Engineering (CPE-POWERENG), 2016, 472-477, DOI: 10.1109/CPE.2016.7544234.

The development of modeling benches for power facilities, which allow solving various functional tasks, is relevant [30].

[30] Gevorkov, L.; Vodovozov, V.; Lehtla, T.; Raud, Z. PLC-based hardware-in-the-loop simulator of a centrifugal pump. Proceedings of  2015 IEEE 5th International Conference on Power Engineering, Energy and Electrical Drives (POWERENG), 2015, 491-496, DOI: 10.1109/PowerEng.2015.7266366.

3) Thank you very much for your comments. We fixed these mistakes.

4) Thank you very much for your comments. We fixed this mistake.

5) Thank you very much for your comments. We fixed these mistakes.

6) Thank you very much for your estimation!

Reviewer 2 Report

(1) The manuscript discusses different loading scheme for gas turbine units, and a mathematical model is built to simulate one loading scheme, comparison with experimental data confirm the correctness of the built model.

(2) The biggest problem of the manuscript lies in its novelty. After reading the whole manuscript, the reviewer cannot locate which part of the work is newly done by the authors.

(3) The English of the manuscript must be examined and revised thoroughly. The reviewer is not a native English speaker, but I did find that the writing inhibits my understanding of the manuscript. Some typos are commented in the attached files for your reference.

(4) Many different loading schemes are introduced in the manuscript, but which one is finally adopted in the simulation and experiment? No clear description was found, until in the result discussion section. This is not natural for readers' understanding.

(5) It seems that only one loading scheme is closely examined, in that case, why were other different loading schemes introduced with a fair large content? A concise introduction of different loading schemes and a detailed description of the one adopted is enough for a research paper.

Author Response

1) Thank you very much for your estimation!

2) We tried to highlight novelty in abstract and discussion:

  • A comparative analysis of the most common methods and technical means of loading the shafts of a free turbine of gas turbine plants intended for operation as part of gas pumping units is presented;
  • Recommendations for the creation of an automation system for load testing of power plants have been developed;
  • Mathematical models and Hardware-In-the Loop-simulation models of power plants have been developed and tested.

Also we provide the formulation of the problem in the chapter 4.1.

3) Thank you very much for your comments! We fixed these mistakes.

4) Thank you very much for your comments.  We added comment about basic loading model in abstract and in chapter 2. Also we added the new links about HIL-simulations methods regarding close investigations.

5) Thank you very much for your comments. To consider the advantages and disadvantages of the basic model "Free Power Turbine Rotor – Hydraulic Brake", it was still necessary to show schemes for the implementation of other models. We tried to make it briefly, since the description of each of the models is quite voluminous.

Round 2

Reviewer 2 Report

The manuscript can be accepted, but a careful proof-reading must be done, particularly in the written English aspect.